# Climate Change Related Catastrophic Rainfall Events and Non-Communicable Respiratory Disease: A Systematic Review of the Literature

**Alexandra M. Peirce** [1,†] **, Leon M. Espira** [2] **and Peter S. Larson** [3,4,*]

1 Medical School, University of Michigan, Ann Arbor, MI 48109, USA; ampeirce@umich.edu
2 Center for Global Health Equity, University of Michigan, Ann Arbor, MI 48109, USA; lespir@umich.edu
3 Social Environment and Health, Institute of Social Research, University of Michigan, Ann Arbor, MI 48109, USA
4 Department of Epidemiology, School of Public Health, University of Michigan, Ann Arbor, MI 48109, USA
* Correspondence: anfangen@umich.edu
† Current address: 7300 Medical Science Building I—A Wing 1301 Catherine St., Ann Arbor, MI 48109, USA

**Abstract:** Climate change is increasing the frequency and intensity of extreme precipitation events, the impacts of which disproportionately impact urban populations. Pluvial flooding and flooding related sewer backups are thought to result in an increase in potentially hazardous human-pathogen encounters. However, the extent and nature of associations between flooding events and non-communicable respiratory diseases such as chronic bronchitis, asthma, and chronic obstructive pulmonary disease (COPD) are not well understood. This research seeks to characterize the state of research on flooding and NCRDs through a systematic review of the scientific literature. We conducted a systematic search of PubMed, Web of Science, and Scopus for published scholarly research papers using the terms flooding, monsoon, and tropical storm with terms for common NCRDs such as asthma, COPD, and chronic bronchitis. Papers were included if they covered research studies on individuals with defined outcomes of flooding events. We excluded review papers, case studies, and opinion pieces. We retrieved 200 articles from PubMed, 268 from Web of Science and 203 from Scopus which comprised 345 unique papers. An initial review of abstracts yielded 38 candidate papers. A full text review of each left 16 papers which were included for the review. All papers except for one found a significant association between a severe weather event and increased risk for at least one of the NCRDs included in this research. Our findings further suggest that extreme weather events may worsen pre-existing respiratory conditions and increase the risk of development of asthma. Future work should focus on more precisely defining measure of health outcomes using validated tools to describe asthma and COPD exacerbations. Research efforts should also work to collect granular data on patients' health status and family history and assess possible confounding and mediating factors such as neighborhood water mitigation infrastructure, housing conditions, pollen counts, and other environmental variables.

**Keywords:** flooding; extreme precipitation; climate change; respiratory health; COPD; asthma

## 1. Introduction

Evidence indicates that climate change-related weather events will become more frequent, last longer, become more intense [1], and have ever-increasing impacts on human health [2,3]. Climate change is resulting in increased temperatures causing more water to be held in the air, increasing the frequency of extreme precipitation events [4]. Floods and intense rainstorms have severe impacts on urban populations; high population densities and fewer porous surfaces to absorb precipitation in urban areas put large numbers of people at risk during severe and sudden rainfall events, particularly marginalized populations [5]. However, the relationships between urban flooding and non-communicable and chronic respiratory conditions are not well understood. This research systematically

reviews the literature on flooding and non-communicable respiratory diseases (NCRDs) such as asthma, chronic bronchitis, and cardiac obstructive pulmonary disease (COPD).

Populated land exposed to climate and flooding risks is expected to increase by greater than 25 percent compared with current levels [6], thus increasing the number and density of people at risk. In the United States, natural hazards and extreme weather events have become more common and increasingly costly to governments and society over the past two decades, a burden of cost that disproportionately impacts cities and urbanized areas [7–10]. Outcomes of flood and extreme precipitation are the result of a confluence of factors which include extreme weather events, the robustness of flood mitigation infrastructure, housing conditions, and individual level vulnerability [11,12], a lack of which is characteristic of many urban areas.

Flooding and extreme precipitation present serious risks to mental and physical health [13–22]. Mental impacts from flooding and extreme precipitation include psychological distress and trauma [23–25]. Physical outcomes are numerous but include gastrointestinal illnesses, skin rashes, infections, and poisoning from exposure to chemical irritants dispersed by flood events [26–32]. Extreme weather and flood events have also been shown to increase transmission of bacterial respiratory infections through inhalation of waterborne bacteria such as Legionella [33] and zoonotic pathogen infections as a result of increased rat populations [34].

The major causes of physical illness from flood events are likely the result of exposure to molds and fungi. Exposure to mold and fungi is associated with the development and exacerbation of NCRDs such as COPD, asthma, and allergic rhinitis (AR) [35,36]. A 2019 review of 27 meta-analyses found that exposure to several types of fungi species *(Cladosporium, Alternaria*, and *Penicillium* spp.) increased the frequency of asthma exacerbations and that exposure to mold as an infant increased the risk of asthma and AR later in childhood [37,38]. Flooding can increase the growth of molds and fungi, which may exacerbate and cause asthma in children [39,40]. Evidence from New Orleans, LA following Hurricane Katrina suggests that homes that experienced flooding were more likely to develop molds and fungus [41]. Home dampness has been found to increase the severity of symptoms for a number of respiratory conditions, including asthma and chronic bronchitis, and to increase exposure to numerous types of microbial agents [42–44]. However, the literature on mold and respiratory health is conflicting results [45], with an EPA fact sheet on molds, pointed out that no real evidence links molds with respiratory illness [46].

The focus of this research is therefore to explore the state of research on associations between flooding, extreme precipitation, and non-communicable respiratory disease (NCRD). We conducted a systematic review of the published literature as available through three popular indexing services.

## 2. Materials and Methods

We conducted a systematic search of the literature using the popular research databases PubMed, Web of Science, and Scopus. We searched all abstracts using terms for catastrophic precipitation events including hurricane, monsoon, flood, and tropical storm. We also included keywords for NCRDs including asthma, AR, bronchitis, and COPD. Database specific search terms can be found in Table 1.

### 2.1. Inclusion and Exclusion Criteria

Only peer-reviewed research papers written in English were considered for inclusion. News reports, review papers, conference abstracts, and communications were excluded from the analysis. This review considered only papers written in English. Case studies, pharmacological, programmatic, experimental, and qualitative studies were excluded. Papers published prior to the year 1990 were excluded. The following study designs were included: randomized controlled trials, cohort, cross-sectional, case-controls, and pre-post exposure comparisons. The article search was conducted between 18 December and 23 December 2021. We only included studies that tested associations between

exposure to a documented extreme weather event (flood, hurricane, etc.) and a specific non-communicable respiratory disease outcome, either through pre- and post-comparisons or by comparisons between affected/not affected populations or regions.

**Table 1.** Search terms and search strategy for each of the three databases used for this paper.

| Search Field | PubMed | Web of Science | Scopus |
|---|---|---|---|
| 1 | (flood OR hurricane OR monsoon OR typhoon OR "tropical storm") | (AB = flood OR AB = hurricane OR AB = monsoon OR AB = typhoon OR AB = "tropical storm") | ABS (flood OR hurricane OR monsoon OR typhoon OR "tropical storm") |
| 2 | (asthma OR "allergic rhinitis" OR "bronchitis" OR "COPD" OR "chronic obstructive pulmonary disease") | (AB= asthma OR AB = "allergic rhinitis" OR AB = "bronchitis" OR AB = "COPD" OR AB = "chronic obstructive pulmonary disease") | ABS (asthma OR "allergic rhinitis" OR "bronchitis" OR "COPD" OR "chronic obstructive pulmonary disease") |

### 2.2. Selection Process

All citations were downloaded to a reference manager service (Zotero). Articles were reviewed and assessed separately by a pair of reviewers in an effort to reduce bias. Each reviewer evaluated the articles to determine if the inclusion criteria were met from the title and abstract. Papers that met the initial criteria were then fully reviewed to determine if they met the inclusion criteria.

### 2.3. Data Collection Process

Once articles had been determined to meet the specified criteria, their study characteristics and findings were recorded in a Google Sheets document. Specific aspects (exposures, outcomes, the population of interest, study length, study design, etc.) of each study were recorded as keywords and included. The team then discussed each selected paper individually and confirmed their suitability for inclusion.

### 2.4. Study Risk of Bias Assessment

We used the Appraisal tool for Cross-Sectional Studies (AXIS tool) to assess the research quality and internal validity of each relevant study [47]. Case-control and cohort studies were evaluated using the eight-item Newcastle–Ottawa Scale [48]. Cross-sectional studies were scored on a 20-point scale, while case-control and cohort studies were scored on a 10-point scale, based on the respective assessment tools used.

## 3. Results

An initial search retrieved 200 articles from PubMed, 268 from Web of Science, and 203 from Scopus for a total of 671 papers. Among these, there were 345 unique papers. Out of this set of unique papers, 89 non-human studies were excluded. These included reviews (17), case reports (3), book chapters (2), letters to editors/conference proceedings (5), and clinical guidelines (6). Twenty-six pharmacologic studies were also excluded. A total of 25 papers were excluded for having a non-respiratory health outcome, and 43 were excluded for lacking a weather-related exposure variable. Of papers that included a weather-related exposure and a respiratory health outcome, 79 were excluded as they studied a non-relevant exposure event (thunderstorm, wildfire, pollution). This left 38 potential candidate papers. A full text review of each resulted in 16 papers being included in the review. Papers spanned the globe including countries such as the US (13), Taiwan (1), and Japan (3). The sample size ranged from 58 to 715,233. Search strategy can be found in Figure 1.

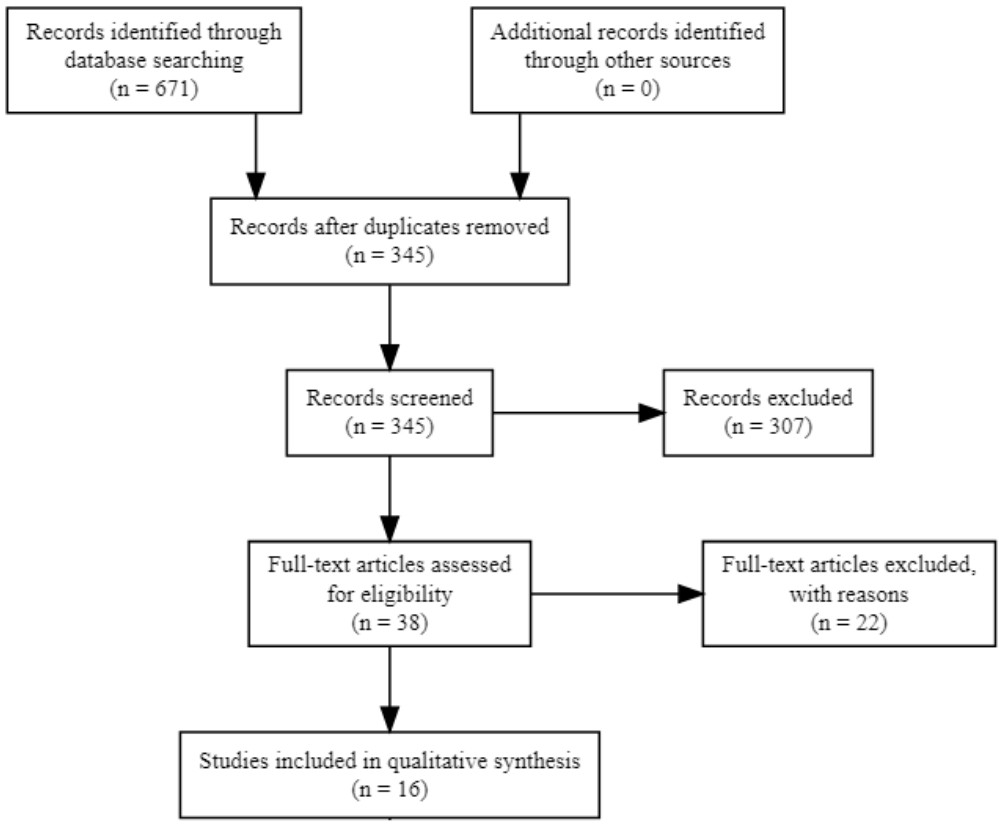

**Figure 1.** Results of search process: PRISMA flow diagram.

### 3.1. Study Characteristics

Study design can be found in Table 2 and results of individual studies are shown in Table 3.

**Table 2.** Study location, duration, sample size and study design for all included publications.

| Author (Year) | Location | Study Duration | Sample Size | Study Design |
|---|---|---|---|---|
| Schinasi LH, 2020 | Philadelphia PA, USA | 2011–2016 | 10,343 children | Case-crossover |
| Shih HI, 2020 | Taiwan | 2008–2011 | 715,233 adults with cerebrovascular disease | Population based case-control |
| Larson PS, 2021 | Detroit MI, USA | 2012–2020 | 4803 households | Cross-sectional survey |
| Fanny SA, 2021 | Houston TX, USA | 2016–2017 | 39,514 children | Cross-sectional survey |
| Eiffert S, 2016 | Atlanta GA, USA | June–August 2014 | 507 households | Cross-sectional survey |
| Saporta D, 2017 | New Jersey, USA | 2003–2015 | 200 patients | Cohort |
| Qu Y, 2021 | New York, USA | 2001–2013 | All COPD hospitalizations during study period (unknown number) | Time-series |
| Chowdhury M, 2019 | US Virgin Islands | 2017–2018 | 11,313 ED encounters | Time-series |
| Hoppe KA, 2012 | Iowa, USA | 2008–2009 | 73 households | Retrospective survey |
| Hendrickson LA, 1997 | Hawaii, USA | August–October 1992 | 1584 primary care and ED encounters | Retrospective record review |
| Cowan K, 2021 | North Carolina, USA | 2010–2011 | All ED visits in 100 NC counties | Retrospective record review |
| Azuma K, 2013 | Japan | 2004–2010 | 379 households | Retrospective survey |
| Cummings K, 2008 | NOLA, USA | 2006 | 553 post-hurricane residents | Cross-sectional survey |
| Brokamp C, 2017 | Ohio, USA | 2010–2014 | 21,108 pediatric asthma ED encounters | Case-crossover |
| Rath B, 2011 | NOLA, USA | 2005–2006 | 1243 children and adolescents | Cross-sectional survey |
| Sato S, 2016 | Fukushima, Japan | 2013 | 58 asthma patients | Cross-sectional survey |

**Table 3.** Type of precipitation exposure, NCRD outcome, measured health outcome, statistical significance (where specified) and direction of association between exposure and outcome for each of the included publications.

| Author (Year) | Precipitation Type | NCRD | Health Outcome | Significance | Association |
|---|---|---|---|---|---|
| Schinasi LH, 2020 | Daily heavy precip. [1] | Asthma | Odds of asthma exacerbation in children were 11% higher on heavy precipitation vs. no precipitation days | 95% CI: (1.02–1.21) | ↑ |
| Shih HI, 2020 | Typhoon | COPD and Asthma | Among affected adults, patients with chronic obstructive pulmonary diseases (COPD) and asthma had significantly increased mortality rates | Adjusted HR: 1.7–2.1 | ↑ |
| Larson PS, 2021 | Pluvial flooding | Asthma | Having at least one adult with asthma in the home was associated with flooding | OR 1.42 [95% CI (1.22, 1.64)] | ↑ |
| Fanny SA, 2021 | Hurricane | Asthma | There was a significantly higher amount of pediatric asthma exacerbation ED visits in the month after the hurricane, but this association was not significant when controlling for seasonal trends | aOR: 1.81 [95% CI (1.54–2.14)] | ↑ |
| Eiffert S, 2016 | Pluvial flooding | Asthma | When controlling for smoking status and length of residence, self-reported current asthma was associated with higher ERMI values [2] | aOR: 1.12, [95% CI: (1.01–1.25); two-tailed $p = 0.04$] | ↑ - |
| Saporta D, 2017 | Hurricanes | Asthma | Post-hurricane patients reported more asthma or lower respiratory symptoms than pre-hurricane (39% vs. 25%) | $p < 0.05$ | ↑ |
| Qu Y, 2019 | Major storms | COPD | Higher rates of COPD hospitalization were associated with major storms, which mainly included flooding, thunder, hurricane, snow, ice, and wind across lag 0–6 days | Adjusted RRs ranged from 1.23–1.49, with significant effects on lag days 0–4 [95% CI: (1.05–2.58)] | ↑ |
| Chowdhury M, 2019 | Hurricanes | Asthma | Higher rates of ED visits for asthma in the two month post-hurricane period compared to pre-hurricane (87.6 asthma patients per 1000 ED visits compared to 74.8) | $p < 0.05$ | ↑ |
| Hoppe KA, 2012 | Flooding | Asthma | Residents of flooded homes experienced more asthma symptoms and required an increase in controller medications | aOR 3.77; [95% CI: (2.06–6.92)] and aOR 1.38, [95% CI: (1.01–1.88)] | ↑ |
| Hendrickson LA, 1997 | Hurricane | Asthma | Primary care visits for asthma were increased in the two week period following Hurricane Iniki | RR: 2.8, [95% CI: (1.93–4.09)] | ↑ |
| Cowan K, 2021 | Hurricane | Asthma | Rates of asthma ED visits were similar in counties that received a disaster declaration and counties that did not | Adjusted rate ratio: 1.02 [95% CI: (0.97–1.08)] | - |
| Azuma K, 2015 | Flooding | Respiratory symptoms–cough/rhinorrhea | Residents of water damaged homes had higher rates of respiratory and nasal symptoms one week after flooding | aOR: 4.19 [95% CI: (1.17–15.0) and aOR: 8.15 [95% CI: (2.39–27.8)] | ↑ |
| Cummings K, 2008 | Flooding | LRS/wheeze | Positive association between exposure to water-damaged homes and lower respiratory symptoms (cough/wheeze) | $p < 0.05$ | ↑ |
| Brokamp C, 2017 | Flooding | Asthma | Increased risk per combined sewer overflow (CSO) event for an asthma-related ED visit was observed 1 and 5 days following CSO events | OR: 1.11 [95% CI: (0.98,1.25)] and OR: 1.12 [95% CI: (0.99,1.27)], respectively | ↑ |
| Rath B, 2011 | Hurricane | Asthma | Self-reported URS and LRS (76% and 36%, respectively) were higher after the hurricane than before the hurricane (22% and 9%, respectively) | $p < 0.0001$ | ↑ |
| Sato S, 2016 | Typhoon | Asthma | 29.3% patients reported worsened asthma symptoms and received systemic corticosteroids as rescue medication over the study period of 1 year (47.0% vs. 19.5%) | $p = 0.033$ | ↑ |

[1] Defined as >95th percentile of summertime distribution. [2] Association between flooding and asthma not directly compared. In bivariate analyses, homes with observed mold were more likely to have resident-reported water leaks [OR = 3.03, 95% CI: (1.51–6.08)].

### 3.2. Risk of Bias in Studies

Study scores from the AXIS appraisal tool of cross-sectional studies ranged from 14 to 18 and both reviewers largely agreed on the scoring of studies. The Newcastle assessment of cohort studies rated all the included studies as good quality. The high rating of studies across both assessment tools is also reflective of the tool's rigid and standardized nature ensuring that most studies that are published would be highly rated using these appraisal tools. See Table 4 for full results.

**Table 4.** Cross Sectional Study AXIS Ratings.

| Author (Year) | Location | Rater 1 Score | Rater 2 Score |
|---|---|---|---|
| Larson PS, 2021 | Detroit MI, USA | 18 | 17 |
| Fanny SA, 2021 | Houston TX, USA | 17 | 16 |
| Eiffert S, 2016 | Atlanta GA, USA | 17 | 16 |
| Qu Y, 2021 | New York, USA | 17 | 17 |
| Chowdhury M, 2019 | US Virgin Islands | 16 | 14 |
| Hoppe KA, 2012 | Iowa, USA | 16 | 16 |
| Hendrickson LA, 1997 | Hawaii, USA | 18 | 18 |
| Cowan K, 2021 | North Carolina, USA | 15 | 15 |
| Azuma K, 2013 | Japan | 17 | 17 |
| Cummings K, 2008 | NOLA, USA | 18 | 17 |
| Rath B, 2011 | NOLA, USA | 16 | 16 |
| Sato S, 2016 | Fukushima, Japan | 18 | 17 |
| Cross-sectional studies | | | |
| Schinasi LH, 2020 | Philadelphia PA, USA | Good quality | Good quality |
| Shih HI, 2020 | Taiwan | Good quality | Good quality |
| Saporta D, 2017 | New Jersey, USA | Good Quality | Good quality |
| Brokamp C, 2017 | Ohio, USA | Good quality | Good quality |

### 3.3. Outcomes and Study Populations

Three papers used asthma prevalence as an outcome [49–51]. Larson et al. found that having at least one adult with asthma in the home was significantly associated with flooding with an OR of 1.42 [95% CI (1.22, 1.64)] [49]. Eiffert et al. found that when controlling for smoking status and length of residence, self-reported current asthma was associated with higher Environmental Relative Moldiness Index (ERMI) values with an aOR of 1.12, [95% CI: (1.01–1.25); two-tailed P=0.04]. While an association between flooding and asthma was not directly compared, on bivariate analyses, homes with observed mold were more likely to have resident-reported water leaks [OR = 3.03, 95% CI: (1.51–6.08)] [50]. Saporta et al. found that post-hurricane patients reported more asthma or lower respiratory symptoms than pre-hurricane (39% vs. 25%, $p < 0.05$) [51], but did not differentiate between asthma prevalence versus symptoms.

Four papers used a clinically significant asthma exacerbation as an outcome [52–56]. Schinasi et al. found that odds of asthma exacerbation in children were 11% higher on heavy precipitation versus no precipitation days with 95% CI: (1.02–1.21) [52]. They defined heavy precipitation days as days with >95th percentile precipitation. One study found that there was a significantly higher amount of pediatric ED visits for asthma exacerbations visits the month after a hurricane with aOR 1.81 [95% CI (1.54–2.14)], but this association was not significant when controlling for seasonal trends [53]. Chowdhury et al. found that there were higher rates of ED visits for asthma in the two-month post-hurricane period compared to pre-hurricane (87.6 asthma patients per 1000 ED visits compared to 74.8, $p < 0.05$) [54]. A study looking at flooding secondary to combined sewer overflow (CSO) events found an increased risk for asthma-related ED visits 1 and 5 days following CSO events, with OR: 1.11 [95% CI: (0.98,1.25)] and OR: 1.12 [95% CI: (0.99,1.27)], respectively [55]. Cowan et al. found that rates of asthma ED visits were similar in counties that received a disaster declaration and counties that did not [56].

Seven studies used participant reported asthma symptoms such as wheezing, chest tightness, or shortness of breath [51,57–62]. A study looking at home flooding found that residents of flooded homes experienced more asthma symptoms and required an increase in controller medications with aOR 3.77; [95% CI: (2.06–6.92)] and aOR 1.38, [95% CI: (1.01–1.88)], respectively [57]. Hendrickson et al. found that primary care visits for asthma symptoms were increased in the two-week period following Hurricane Iniki with RR: 2.8, [95% CI: (1.93–4.09)] [58]. One study found that residents of water-damaged homes had higher rates of respiratory and nasal symptoms one week after flooding with aOR: 4.19 [95% CI: (1.17–15.0) and aOR: 8.15 [95% CI: (2.39–27.8)], respectively [59]. Cummings et al. found a positive association between exposure to water-damaged homes and lower respiratory symptoms such as cough and wheeze with $p < 0.05$ [60]. Rath et al. found that self-reported upper-respiratory symptoms (nasal congestion) and lower-respiratory symptoms (coughing and wheezing) were higher after a hurricane (76% and 36%, respectively) than before the hurricane (22% and 9%, respectively) with $p < 0.0001$ [61]. Another study demonstrated that 9.3% of their participants reported worsened asthma symptoms after a typhoon, and an increased proportion of participants received systemic corticosteroids as a rescue medication over the study period of 1 year (47.0% vs. 19.5%, $p = 0.033$) [62]. Of note, it is standard of care to treat an asthma exacerbation with systemic corticosteroids, but data on asthma exacerbation frequency were not collected in this study.

Two papers studied COPD. Qu et al. used COPD exacerbations requiring hospitalization as their outcome and found that higher rates of COPD hospitalization were associated with major storms, which mainly included flooding, thunder, hurricane, snow, ice, and wind across a lag of 0–6 days. Adjusted RRs ranged from 1.23–1.49, with significant effects on lag days 0–4 [95% CI: (1.05–2.58)] [63]. Shih et al. found that among adults with cerebrovascular and cardiac disease, patients with chronic obstructive pulmonary diseases (COPD) and asthma had significantly increased mortality rates after a typhoon, with adjusted HR 1.7–2.1 [64].

No studies on AR that fit the search criteria were found. Papers reviewed for inclusion on AR had average daily precipitation as their exposure available [65], which did fit the criteria of extreme weather.

Study population ages and locations were varied among papers. Four papers studied children [52,53,55,61]. Brokamp et al. defined cases as aged 0–18 years who visited the Cincinnati Children's Hospital Medical Center between January 2010 and December 2014. Fanny et al. selected individuals under the age of 18 who were seen at pediatric emergency departments and urgent care centers in Houston, Texas one year before and one month after Hurricane Harvey. Rath et al. included children and adolescents below the age of 24 residing in New Orleans immediately following Hurricane Katrina; 90% of their sample was age 15 or below [61]. Schinasi et al. studied the population of children in Philadelphia, PA aged 0–18 years.

Four studies included children and adults [49,51,56,58]. Cowan et al. included all patients who visited a hospital for asthma after Hurricane Irene, ranging from age 1 to 101; 44% of participants were under the age of 18 [56]. Hendrickson et al. included all Kauai residents who sought outpatient, inpatient, or emergency care after Hurricane Iniki. The study population in Larson et al. was comprised of Detroit, MI households. Saporta et al. sought to represent the population of Northern New Jersey impacted by Hurricane Irene and Sandy. Their pre-hurricane group and post-hurricane group had 12% and 29% of participants below the age of 18, respectively [51].

Six papers studied adults [50,57,59,60,62–64]. Azuma et al. chose adults who experienced flooding secondary to typhoons or heavy rainfall in six Japanese cities between 2004 and 2010. Chowdhury et al. studied the population of St. Thomas in the US Virgin Islands impacted by Hurricane Irma and Maria and did not specify whether they included an adult or pediatric population. Cummings et al. included adults residing in New Orleans in March 2006. Eiffert et al. selected adults living in the English Avenue and Vine City neighborhoods of Atlanta in the Proctor Creek watershed. Hoppe et al. included individu-

als affected by the 2008 Cedar Rapids flood in Iowa. Qu et al. selected adults in New York State hospitalized with a primary diagnosis of COPD. Sato et al. sought to study asthma patients in Japan during typhoon season; while their study population was not directly specified, at least 95% of their sample was above the age of 18 based on mean and standard deviation [62]. Shih et al. studied the adult population of Taiwan affected by Typhoon Morakot who carried a diagnosis of acute ischemic heart disease, intracranial hemorrhage, or ischemic stroke.

One study in Japan on typhoons and asthma exacerbations studied a population of persons with asthma who were currently undergoing treatment at the time of study [62]. They excluded patients who had upper or lower respiratory infections or who had changed their treatment regimen during the study period.

### 3.4. Case Definitions

Asthma exacerbations are defined by having reduced expiratory airflow function along with one or several of the following symptoms: shortness of breath, cough, wheezing, or chest tightness [66]. The Fuhlbrigge reference appears to be a standard definition. One paper used this definition for their cases [52].

However, many papers did not use a standard definition, and simply asked participants to self-report asthma symptoms. Some papers [59,60] asked participants to report their asthma symptoms in an interview or questionnaire. Other papers [50,57] asked participants to report any asthma symptoms and whether they had received a diagnosis of asthma from a doctor in the past. Larson et al. asked participants if they or members of their household carried a diagnosis of asthma [49]. Sato et al. conducted a patient interview, and recorded data on participants' asthma symptoms and whether they had needed a systemic corticosteroid as a rescue medication (a proxy for asthma exacerbation). They defined worsening symptoms as "a condition where a subject has become aware of at least one symptom change or took reliever medication" [62]. One study used data from the Health Survey for Children and Adolescents After Katrina survey and examined self-reported upper and lower respiratory symptoms, asthma diagnosis, and asthma attacks [61]. Saporta et al. conducted a chart review of patients reporting asthma symptoms [51].

Other papers conducted a record review using ICD codes. For studies using asthma as a health outcome, most used the ICD-9 493.XX definition, while others used the ICD-10 system with J45 as their diagnosis code. Brokamp et al. defined cases as ED visits with a primary diagnosis using ICD-9 [55], while Chowdhury et al. defined ED visits with the ICD-10 classification [54]. One paper used both the ICD-9 and ICD-10 systems [52]. Cowan et al. conducted a record review of ED visits and inpatient visits with asthma as the primary diagnosis using ICD-9 [56]. Fanny et al. also reviewed ED cases and stated they used ICD-9 and ICD-10 systems to identify asthma cases, but did not specify exact codes [53]. Another study reviewed records from EDs, inpatient admissions, and outpatient clinics in their area with a primary diagnosis of asthma using ICD-9 [58]. Shih et al. reviewed both outpatient and inpatient records and defined asthma cases using ICD-9, and also defined COPD cases using the ICD-9 codes 491–492, 494, and 496. They counted mortality cases if a patient's record was withdrawn from the Taiwan NHI insurance database due to death [64]. One study using COPD as their health outcome reviewed inpatient records and chose cases where the COPD ICD-9 codes 491, 492, and 496 were listed as the principal diagnosis for admission [63].

### 3.5. Study Design

One paper used a time-stratified case crossover design of associations of health precipitation and pediatric asthma exacerbation events in Philadelphia. Control days were defined as those falling on the same day of the week within the same month and year and the exacerbation, examining effects by season [52].

Both studies from Japan used retrospective study designs. The first selected patients receiving treatment and then administered a survey asking patients to recall outcomes of

asthma symptoms and medicinal use after a flooding event [62]. The other selected six different clinic sites throughout Japan and then selected households within the catchments of those clinics. The selection criteria for households were unclear. Households were provided with survey forms, which they were to fill out and return by post [59].

Several papers conducted retrospective analyses of catastrophic precipitation events on records of ED visits. One study from North Carolina looked at ED visits for asthma concerns (ICD-9 493.XX) using data from the Healthcare Cost and Utilization Project State Emergency Department Database and the State Inpatient Database for the years 2010 and 2011 [56]. Home addresses were used to assess exposure. Investigators compared counties exposed to Hurricane Irene, identified through FEMA classification of major disasters with two counties that were not.

### 3.6. Statistical Analysis

Sato et al. had a small sample size and conducted simple Chi-square, t-tests, and Fisher's exact tests for differences in exposure variables such as pollen concentrations between persons whose symptoms worsened following typhoons and those whose symptoms did not worsen [62]. They also compared these groups using a logistic regression model including multiple measures of IgE antibodies and rescue us of systemic CS for the past year.

Another study from Japan compared the health impacts of flooded and non-flooded homes using Chi-square tests for bivariate associations. Researchers used multivariate models to compare flooded and non-flooded homes controlling for sex, age, and household factors. Though the study was conducted by sampling from several different sites, no attempt was made to include regional random effects [59].

The study of asthma-related ED visits from North Carolina compared counties impacted by Hurricane Irene with those that were not. They used a difference of differences analysis conducted on a log scale using a Poisson generalized linear model to estimate the impact of the hurricane on counts of ED visits. This model controlled for the count, month, and year. They took spatial factors into account (correlation between counties) using an autoregressive-1 correlation structure [56].

Some studies used systematic sampling strategies. A study from New Orleans selected residential census tracts at random and then randomly assigned waypoints within selected tracts to select households. When persons refused or were not home, teams went to another household using an unspecified criterion [60].

The study from New Orleans created a scoring system for both exposures and outcomes. Exposures were measured through a "clean up score" equal to the sum of the reported number of homes cleaned that had less than 50 percent of walls and ceilings covered with mold. From there, they categorized this score along with other contextual information. For outcomes, they used a composite score of asthma symptoms as measured by the survey instrument [60].

### 3.7. Exposure Assessment

For extreme storm exposures, several methods were used. Schisani et al. defined extreme precipitation as storms with greater than the 95th percentile of precipitation for the year using data from NOAA and controlling for temperature [52]. Qu et al. obtained major storm dates and locations obtained from the Integrated Surface Database of the U.S. National Oceanic and Atmospheric Administration [63].

Chowdhury et al. studied hurricanes as their exposure variable. The ED evaluated in the study is located in the only hospital on St. Thomas, and it was assumed all patients presenting to the ED were affected by hurricanes Irma and Maria [54]. An additional paper with an island population assumed all residents of Kauai were exposed to Hurricane Iniki [58]. Another paper studying hurricanes took patients' billing addresses from ED records and considered patients who lived in a county that received a FEMA disaster declaration from Hurricane Irma as exposed [56]. Fanny et al. assumed the entire popu-

lation of Houston was exposed to Hurricane Harvey as one-third of the city flooded but did not collect individual exposure data [53]. Another paper chose their non-exposed vs. exposed group by identifying patients in the same clinic who were evaluated before and after Hurricane Harvey, respectively [51].

Sato et al. assumed all Fukushima residents were exposed to the 2013 typhoons [62]. Another paper chose patients for their study who were designated as living in an affected area of Typhoon Morakot by the Taiwan National Health Insurance (NHI) Database [64]. The NHI had identified persons living in affected areas in order to provide financial help for their medical expenses.

Papers that identified flooding as their extreme weather event used similar methods. Azuma et al. obtained information from the Tokyo Ministry of Land, Infrastructure, Transport and Tourism to identify severely flooded areas, and then worked with local public health centers to identify households with water-damaged homes [59]. Another paper obtained the locations and dates of combined sewer overflow events from the Metropolitan Sewer District (MSD) of Greater Cincinnati; a household was considered exposed if it was within 500 m of an overflow flood event [55]. Hoppe et al. designated homes as water-damaged by the Cedar Rapids flood if they were marked for follow-up by the Code Enforcement Division of Cedar Rapids [57].

Other papers used participant questionnaires to determine exposure to flooding. Cummings et al. conducted interviews on water damage including the extent of walls and ceilings covered in mold [60]. A study on urban pluvial flooding identified houses where the house had flooded from the outside with water covering at least a quarter of the floor of a room [50]. A similar study used participant questionnaires on whether flooding had occurred in the home since they had lived there [49]. Rath et al. used participant-reported flood damage as their exposure assessment [61].

*3.8. Results of Individual Studies*

Of the 16 papers that were selected for review, weather events included flooding, heavy precipitation, hurricane, and typhoon. Non-communicable respiratory conditions included asthma and COPD.

For studies evaluating a flooding event, six papers found significant association between home flooding and asthma [49,50,55,57,59,60]. Qu et al. [63] found a significant association between major storms, which included flooding, and COPD. However, this study did not differentiate between different types of storms in their analysis.

Four papers found significant associations between hurricanes and asthma symptoms [51,54,58,61]. Qu et al. found an association between COPD hospitalizations and major storms but did not differentiate the type of storm in analysis [63]. Cowan et al. found no association between asthma ED visits in counties affected by a hurricane (counties that received a disaster declaration) and those who did not [56]. Fanny et al. found a significant association between hurricanes and pediatric asthma exacerbations, but this association was not significant when accounting for seasonal trends [53].

One study a found significant association between typhoons and asthma exacerbations requiring steroids [62]. However, this study had a small sample size and did not adequately control for confounding factors in their analysis.

## 4. Discussion

The papers selected for this literature review involved cohort and case control studies examining the effect of hurricanes, typhoons, flooding, and severe storms on non-communicable respiratory diseases, in children and adults. All sixteen papers except for one [56] found a significant association between a severe weather event and worsening or prevalence of non-communicable respiratory diseases. Most studies examined asthma as their health outcome. For asthma, three studies suggested a relationship between severe weather and asthma prevalence. Three studies suggested a relationship between asthma exacerbations, while one did not. Seven studies suggest an association between severe

weather and asthma symptoms not specifically classified as an exacerbation. Two studies were about COPD. One showed a relationship between COPD exacerbations and severe weather. Another found that patients with COPD had higher mortality rates after a typhoon but did not specifically study rates of diagnosis or exacerbations [67]. Typhoons have been associated with the development of allergic sensitization in patients with asthma [68]. Although AR was included in the search criteria, no qualifying studies on the development of allergic rhinitis (rather than simple sensitization) were found.

Our findings suggest that extreme weather events may worsen preexisting respiratory conditions and increase the risk of developing asthma. Severe storms such as hurricanes and typhoons can cause coastal flooding, and overwhelm city infrastructure causing sewer overflows and pluvial flooding. Resultant building water damage can encourage the growth of mold and fungus, worsening indoor air quality. Though few, previous studies have suggested that the putative causal relationship between flooding and asthma is the result of the deterioration of indoor air quality because of mold. A report by the World Health Organization stated that there is a concrete relationship between indoor dampness and mold and childhood asthma [69]. Furthermore, a review of 45 primary literature studies and three meta-analyses including asthma, AR, and respiratory infections concluded that there was a relationship between indoor dampness and the development of asthma and worsening of previously diagnosed respiratory conditions [70,71]. Given that even in the United States, a significant number of homes are affected by mold and dampness, both of which may aggravate existing asthma or provoke the development of asthma, further structural damage from climate change such as flooding and sea-level rise will likely further worsen indoor air quality [19]. However, the literature is not unanimous on the association between non-communicable respiratory disease and extreme weather events. For example, no clear causal relationship has been established between damp indoor spaces and health cite [45]. No association between mold and respiratory health was detected in the aftermath of Hurricane Katrina in one study [72].

Getting a clearer understanding of the causal relationship between flooding and respiratory outcomes such as COPD and asthma is further hampered by the difficulty in obtaining a validated diagnosis and by the complexity of the causal pathway from water damage, mold, and respiratory outcomes. A review of 16 studies indicated that the risk for asthma increased along a causal pathway of water damage, dampness, visible mold and mold odor, however clear delineating pathophysiological pathways is a challenge [73]. Among possible pathways is that $\beta$ -glucans and chitins present in fungal cell walls could promote airway epithelial cell hyper-responsiveness and inflammation through a Th2- and Th17-mediated immune response [37,74]. It has also been shown that fungal proteases directly induce airway epithelial cell inflammation by disrupting smooth muscle cell extracellular matrix cell interactions, inhibiting mucociliary clearance, and activating the innate immune system [74]. Patients with allergic asthma can become sensitized to fungal allergens through an IgE-mediated response [75]. In addition, fungi exposure has been shown to cause rare conditions such as allergic bronchopulmonary aspergillosis through an eosinophilic Type I hypersensitivity reaction, and hypersensitivity pneumonitis through a cell-mediated Type IV hypersensitivity reaction [76]. Mapping out this association will require that more studies use validated outcome measures such as pulmonary function testing and not rely on mail surveys or in-person interviews. However, many of the papers in this review relied on self-reported health measures that were not validated by medical records or by health professionals. Only one study used a validated tool to define exacerbations [52].

There is also an inherent difficulty in studying environmental exposures. For example, every study defined flood exposure in a different way, complicating cross-study comparisons. There are also often multiple confounders present. Some studies controlled for home quality [49] which can affect the susceptibility of a home to flooding, while others did not. Other potential confounders include environmental factors that could have been present and that are known to affect respiratory conditions such as heat and pollen concentrations.

While Sato et al. controlled for pollen conditions in their analysis, other studies did not [62]. In addition, there are other significant environmental risk factors in the development of asthma including family history, tobacco use in the home, pollution, prior respiratory infections, and obesity [77]. No study controlled for all of these environmental factors.

This review further adds to the body of evidence suggesting an association between extreme weather events and non-communicable respiratory disease. Future work should focus on accurately defining health outcomes measures using validated tools to describe asthma and COPD exacerbations [66,78] if relying on self-reported health data. There should also be a more concerted effort to collect data on family history and control for confounding factors such as home quality, pollen counts, and other environmental variables. This is especially important from an environmental justice standpoint. Rates of COPD and asthma are higher in historically marginalized communities [79,80], who will be impacted more severely by severe storms as they lack the resources to relocate during severe storms and protect their homes from flood waters. Future work should strive to prioritize low-income and communities of color. There are however gaps in the review. While we had hoped to include studies on AR, none were found that fit our search criteria. We could have widened the criteria to include studies on this condition as it will likely be affected by climate change as it is affected by changes in atmospheric pollen and precipitation [65]. Future work could include how AR is affected by weather conditions outside of extreme storms.

Finally, this work indicates that flooding and EWE present significant threats to respiratory health. This would indicate that disaster prevention and relief efforts that focus on flooding and EWE include respiratory health along with waterborne disease, injuries, or mental health. Governments and communities should bolster efforts to raise awareness of respiratory risks are bolster efforts to prevent the worst impacts of flooding both before and after the event. Preparation could include supporting programs that subsidize home improvements for low-income homeowners and landlords who rent to low-income residents. Research work in the US has indicated that housing condition is an important contributor to flood risk and water in basements [49]. Improvements to homes would prevent water from entering homes, thereby preventing the development of microbes and indoor contaminants. Similarly, programs that support abatement of moisture (e.g., FEMA programs and Small Business Administration disaster loans in the US) in the home after a flood or EWE could prioritize low-income residents to minimize the risk of indoor exposures to microbes.

## 5. Conclusions

The impacts of climate change will differ between urban and rural areas. Each locale will face unique challenges and risks, with risks in urban areas likely modified or driven by the built environment. Most of the studies in the review were conducted in urban areas. They all highlight that cities may be disproportionately vulnerable to flood events, placing their residents at high risk for flood-associated respiratory disease. As shown by the studies conducted in Detroit [49] and Houston [51,53] infrastructure investment and improved water management systems could be especially important disease mitigation measures. Home air quality is also often an overlooked exposure, which may however grow in importance as the effects of climate change manifest. A focus of future public health campaigns could be on improving home quality to treat underlying causes of asthma, which has been shown to improve disease control and healthcare costs in asthmatic children [81,82]. Further work is therefore needed to determine at which level—city or household—should interventions focus and which will have the biggest impact on public health outcomes.

**Author Contributions:** Conceptualization, A.M.P. and P.S.L.; methodology, P.S.L.; formal analysis, A.M.P., P.S.L. and L.M.E.; data curation, A.M.P.; writing—original draft preparation, A.M.P., P.S.L. and L.M.E.; writing—review and editing, A.M.P., P.S.L. and L.M.E. All authors have read and agreed to the published version of the manuscript.

**Funding:** This research was conducted in part through support from the University of Michigan Institute for Global Biological Change, the University of Michigan Center for Global Health Equity and the University of Michigan Medical School.

**Institutional Review Board Statement:** Not applicable.

**Informed Consent Statement:** Not applicable.

**Data Availability Statement:** Not applicable.

**Acknowledgments:** The authors would like to acknowledge Carina Gronlund of the University of Michigan Institute for Social Research and Toby Lewis of the University of Michigan Medical School.

**Conflicts of Interest:** The authors declare no conflict of interest.

## Abbreviations

The following abbreviations are used in this manuscript:

| | |
|---|---|
| AR | allergic rhinitis |
| AXIS | Appraisal tool for Cross-Sectional Studies |
| CI | Confidence interval |
| COPD | chronic obstructive pulmonary disease |
| ED | Emergency department |
| HR | Hazard ratio |
| ICD | International Classification of Disease |
| NCRD | non-communicable respiratory disease |
| OR | Odds ratio |
| RR | Relative risk |

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
