# Peer review of "Climate Change Related Catastrophic Rainfall Events and Non-Communicable Respiratory Disease: A Systematic Review of the Literature"

_climate, doi:10.3390/cli10070101_

Round 1

Reviewer 1 Report

The study presents an interesting systematic review of the linkages between catastrophic rainfall events and non-communicable respiratory disease. It seems to be well done on the whole. I have some major comments and minor comments, as set out below. It adds to the growing literature on climate and health – bringing focus on a little considered issue.

Major comments

1.  One concern I have in the focus on extreme rainfall is the potential for cross-over with other climatic events – e.g. thunder/lightning storms that may go ahead of heavy rain and if may be difficult to disentangle the influence of such conditions from the rainfall that follows (or goes alongside). There is recent literature on e.g. the influence of thunderstorms on asthma – which I would imagine would not necessarily be picked up by this search (but might precipitate some of these events) – e.g. work by Lindstrom and Silver (Lindstrom et al, 2017; Silver et al, 2018).  It might be worth a sentence or two on what you are not looking at – and how co-occurrence may affect the findings potentially.

2. Please check the numbers in Figure 1 as the number of articles found in this (9) does not correspond with the 16 papers included in the review at the end.  Were any studies found through searching references in the found articles? It seems strange to me that there were none found through other sources – and if this is the case I’m not sure why you include the box at the top of Figure 1.

43. I am surprised you didn’t come up with any study on AR and floods – this may be due to your inclusion/exclusion criteria (which might need to be more detailed than it is). E.g. Visitsunthorn et al (2018) find changes in sensitivity for asthmatics and for those with AR for those exposed to flooding in Bangkok. This may have been excluded because of the health outcome measure – but you may want to tighten up your description in the methods.

54. Section 3.2 seems to be missing a lot of text! (Table 4 should be cited somewhere – and indeed other tables in the text if they are not)

65. I think the text in Section 3.9  should be in the methodology – and perhaps Section 3.2 (the results of the appraisal) should be moved to become Section 3.8. It is interesting that you find that all studies rate pretty well – is this not unusual?

76. For the discussion section -  I note that since you wrote this, there may have been new studies on parts of your work – e.g. Cowan et al (2021) on hurricanes and asthma in children. It might be good to quickly update to ensure you aren’t missing some of these.

87. On the further work – I wonder if anything you came across looked at sensing the indoor air (and linking to health outcomes) in the contexts mentioned. This might be useful. And I wonder also about how your findings may influence disaster response – the focus is often on water borne disease or disaster related injuries or mental health. Is there need for increased awareness among practitioners of the risks for populations from non-communicable respiratory disease?

Minor comments

1. I have printed the file out – and I have to say that I think the font in the Tables is very small. You may want to consider dividing them across pages to allow the font to be bigger or cutting into smaller tables. 

2. Line 73 – change topical to tropical

3. Table 3 – there is a 1 on the second row of the second column that seems not to link to anything – perhaps it is meant to be linked to a note?

4. References 68,77 and 78  - seems odd to have a publisher in a citation for a journal.

References

Cowan et al (2021) Impact of Hurricanes on Children with Asthma: A Systematic Literature Review. Disaster Medicine and Public Health Preparedness.  16(2), 777-782. doi:10.1017/dmp.2020.424

Lindstrom et al (2017) Thunderstorm asthma outbreak of November 2016: a natural disaster requiring planning. Medical Journal of Australia Vol 207, Issue 6  doi: 10.5694/mja17.00285

Silver et al (2018) Seasonal asthma in Melbourne, Australia, and some observations on the occurrence of thunderstorm asthma and its predictability. PLOS ONE E 13(4): e0194929. https://doi.org/10.1371/ journal.pone.0194929

Visitsunthorn, N. et al (2018) Great flood and aeroallergen sensitization in children with asthma and/or allergic rhinitis Asian Pac J Allergy Immunol. 2018 Jun;36(2):69-76. doi: 10.12932/AP0886. PMID: 29161054.

Reviewer 2 Report

what is the justification for stating precipitation impacts more urban populations. Last year in USA crops could not be planted-fields too wet. In India and Bangladesh farmers effected more seriously than urban dwellers. A case needs to be made for this statement. line 29- sentence redundant.  Small number of papers and different diseases could not be helped but intuitively one would think more water greater water related illness. 

Author Response

Please see attachment. Review #2 comments are second. 

Round 2

Reviewer 2 Report

none